# Astrocyte diversity and subtypes: aligning transcriptomics with multimodal perspectives

Maroussia Hennes (iD)[1,2], Maria L Richter (iD)[1,2], Judith Fischer-Sternjak (iD)[1,2 ✉] & Magdalena Götz (iD)[1,2,3 ✉]

## Abstract

Astrocytes are considered a diverse cell population, carrying out many functions essential for supporting neuronal activity. The surge of sc/snRNA-sequencing data greatly expands our understanding of heterogeneous astrocyte gene expression, but also leads to confusion about the multitude of described astrocyte subtypes and substates in the mammalian brain. Here we discuss and review the definition of distinct subtypes and the evidence for this amongst astrocytes. Determining whether an astrocyte subtype represents a stable identity or a dynamic substate requires generalization of findings across datasets, incorporation of validation, and ideally, functional analyses. How to best achieve this is the focus of this review, including considerations about the different transcriptomic approaches. We further discuss the alignment of astrocyte subtype transcriptomes with other hallmarks, such as their position. These considerations are embedded in an overview of the current astrocyte heterogeneity knowledge as a basis for subtype definitions using different analysis techniques. Following technical and biological considerations of transcriptome analyses, we advocate for multimodal alignment to identify stable astrocyte subtypes.

## Preamble

In the central nervous system, the classical cell type distinction is between neurons and glia. Within a cell type group, subtype definitions have been largely conceptualized by the criteria used for neuronal subtypes, while astrocytes have been considered rather homogenous for a long time. For example, astrocyte-blood vessel contact was presumed common to all, yet this remains unclear. However, in recent years, astrocyte heterogeneity has been observed in various aspects, prompting the question of how to define their subtypes. Single-cell/nuclei transcriptomics has brought further new dynamics to the definition of subtypes or substates by retrieving clusters of astrocytes with heterogeneous gene expression. These studies however vary in methods and criteria for identifying astrocyte subtypes and often lack functional validation. It is therefore timely to provide an overview and consider the evidence for astrocyte heterogeneity. With this review, we aim to critically assess subtype and substate definitions and integrate existing transcriptomic findings within the framework of how to define astrocyte subtypes and to which extent neuronal subtype criteria may also be applicable to astrocyte subtype definitions.

**Keywords** Astrocytes; Heterogeneity; sc/snRNA Transcriptomics; Subtype/State; Multiomics
**Subject Categories** RNA Biology; Neuroscience

## Introduction

### The evolving concept of astrocyte heterogeneity

Astrocytes are one of the most abundant cell types in the central nervous system (CNS), responding to various stimuli through morphological, molecular, and functional adaptations (Schiweck et al, 2018; Barres, 2008; Wahis et al, 2021). Even though their importance for maintaining adequate brain functioning was long underestimated, astrocytes are now well recognized as being indispensable players in ensuring CNS homeostasis in health and disease (Barres, 2008; Verkhratsky et al, 2021, 2023). They fulfill a multitude of significant tasks to preserve the CNS equilibrium such as providing metabolic, antioxidant and trophic support, regulating the potassium balance, maintaining the blood-brain barrier (BBB) and securing synaptic homeostasis (Allen, 2014; Chung et al, 2015; Allen and Eroglu, 2017; McBean, 2018; Verkhratsky and Nedergaard, 2017; Weber and Barros, 2015). Astrocytes also exhibit remarkable plasticity, allowing them to tailor these tasks to physiological or pathological changes in their environment (Patani et al, 2023; Pestana et al, 2020; Zhang and Barres, 2010). However, it is unclear to which extent all astrocytes perform all of these functions, and how heterogeneous and diverse astrocytes really are. Therefore, this review starts with a brief historical overview about astrocytes, the initial evidence for their heterogeneity, and how they align with the available criteria for subtype definition (as depicted in Fig. 1).

Drawing inspiration from the neuronal field, morphology and location are amongst the first criteria for defining subtypes, also because methodologies to study this have been available for some

[1]Chair of Physiological Genomics, Biomedical Center (BMC), Faculty of Medicine, LMU, Munich, Germany. [2]Institute of Stem Cell Research, Helmholtz Center Munich, German Research Center for Environmental Health, Neuherberg, Germany. [3]Excellence Cluster of Systems Neurology (SyNergy), Munich, Germany.
✉E-mail: judith.fischer@helmholtz-munich.de; magdalena.goetz@helmholtz-munich.de

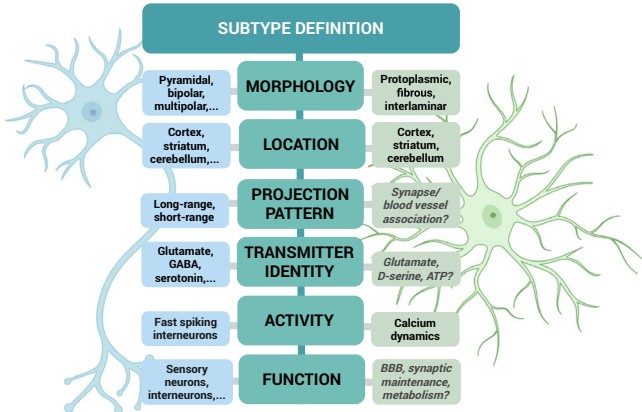

**Figure 1. Extending subtype paradigms from neurons to astrocytes.**

Classification of astrocyte subtypes can benefit from comparison to the neuronal field, where subtype definition is more straightforward. Some neuronal subtype criteria—such as morphology and location—can be readily extrapolated to astrocytes, while others (gray, italic) remain subject to debate. Created in BioRender. Hennes M (2025). https://BioRender.com/1jfaq2g.

time. Already, Cajal showed in his histological preparations and drawings a clear morphological difference between the "protoplasmic" astrocytes in the gray matter (GM) and "fibrous" astrocytes in the white matter (WM) (Garcia-Lopez et al, 2010). Fibrous astrocytes display straight and long processes, whereas protoplasmic astrocytes have highly branched processes that penetrate the neuropil and typically ensheath synapses and blood vessels (Barres, 2008; Khakh and Deneen, 2019). In the human neocortex, protoplasmic astrocytes are even more complex and can be further divided into morphological subclasses, such as the interlaminar and the varicose projection astrocytes (Oberheim et al, 2009).

Given that neurons can be classified by their projection patterns, astrocytes might similarly be distinguished by their ability to contact blood vessels or synapses. As part of the neurovascular unit, astrocyte endfeet can contact blood vessels and contribute to the maintenance of the BBB. Interestingly, astrocyte-vessel contact has been shown to be variable with, for example, astrocytes in deeper cortical layers contacting more vessels (Hösli et al, 2022). How astrocyte-vessel contact changes across regions, however, remains rather unexplored. In addition, an astrocyte subtype with particularly close contact to the blood vessels has been identified as juxtavascular astrocytes (Götz et al, 2021). These astrocytes reside with their somata at the blood vessels and have a clonal origin, i.e., are derived from a common progenitor cell as opposed to the non-juxtavascular astrocytes (Götz et al, 2021). Their channel composition suggests that they also may differ in their function with a bias to resume proliferation after brain injury (Götz et al, 2021; Sirko et al, 2013; Bardehle et al, 2013). On the other hand, astrocytes also exhibit variability in the proximity of their processes to neuronal synapses and in the extent of their synaptic coverage (Chai et al, 2017; Lanjakornsiripan et al, 2018). While regions such as the cortex contain a high proportion of tripartite synapses—where astrocytes interact with pre- and postsynaptic neurons—their prevalence is markedly reduced in the spinal cord (Broadhead et al, 2020; Oberheim et al, 2012; Farhy-Tselnicker and Allen, 2018).

Neurotransmitter identity serves as another clear and widely used criterion for classifying neuronal subtypes. Likewise, during the process known as gliotransmission, astrocytes can release a variety of small molecules—referred to as gliotransmitters—that are capable of modulating synaptic activity (Araque et al, 2014; Savtchouk and Volterra, 2018; Covelo and Araque, 2018). Whether different astrocyte subtypes release distinct gliotransmitters is still under investigation. However, recently De Ceglia et al provided evidence for a subpopulation of highly specialized hippocampal astrocytes that are functionally competent for vesicular glutamate transporter-dependent glutamate release (de Ceglia et al, 2023).

Compared to neurons, astrocyte activity cannot be measured in Na-channel mediated action potentials; instead, they respond to environmental stimuli by producing intracellular $Ca^{2+}$ signals (Bindocci et al, 2017; Semyanov et al, 2020). Heterogeneous astrocyte activity, as measured by $Ca^{2+}$ signaling, has been demonstrated in regions such as the hippocampus and striatum. Astrocytes in these areas show differences in both spontaneous and evoked $Ca^{2+}$ events, indicating the existence of neural circuit-specialized astrocytes (Chai et al, 2017). $Ca^{2+}$ activity in astrocytes varies not only across brain regions but also between distinct cortical layers within the somatosensory cortex (Takata and Hirase, 2008). Despite the conceptual parallels with neurons, the classification of astrocyte subtypes based on their activity is still not fully resolved.

Technological advances in RNA sequencing (RNA-seq), such as single-cell/nuclei (sc/sn)RNA-seq, have opened new avenues to study the molecular diversity of astrocytes in greater depth. Initial studies implementing bulk RNA-seq described molecular differences between astrocytes from different CNS regions (Chai et al, 2017; Boisvert et al, 2018; Clarke et al, 2018; John Lin et al, 2017; Karpf et al, 2022; Morel et al, 2017). Likewise, the use of Mlc1-eGFP mice aimed to characterize the molecular profile of astrocytes in contact with blood vessels (Yosef et al, 2020). The next step was to explore the extent to which astrocytes from the same region differ from one another, potentially revealing additional layers of heterogeneity.

This question could be addressed using sc/snRNA-seq: based on the gene expression profile, astrocytes can be grouped into clusters, differing between as well as within brain regions (Batiuk et al, 2020; Bayraktar et al, 2020; Bocchi et al, 2025; Ohlig et al, 2021; Lanjakornsiripan et al, 2018) (Table 1). Heterogeneity at the transcriptome level also greatly enriched our understanding of astrocytes in pathological conditions such as neurological disorders, inflammation, and brain injury. Interestingly, these injury/disease-induced changes could be either protective or detrimental for disease progression (Schober et al, 2022; Sofroniew, 2020; Endo et al, 2022; Patani et al, 2023; Sadick et al, 2022).

However, how coherent are these data—both in the intact and in the diseased CNS? Are there common signatures and how well are they aligned?

For example, not all the described gene expression clusters can be identified across different datasets. Moreover, it is also not clear to which extent astrocytes clustered by their gene expression indeed correspond to functional or morphological subtypes. Considering how dynamic astrocytes can be, we aim here to discuss to which extent the heterogeneity found by single readouts—morphology, position, expression—may align with function and how best to define a subtype versus a "substate"—a transient state of a cell.

**Table 1. Overview of studies exploring homeostatic astrocyte heterogeneity using sc/snRNA-seq.**

| Study | Methods | Brain region | Species, including age and sex | Number of cells | Astrocyte subclusters |
|---|---|---|---|---|---|
| Zeisel et al, 2015 | scRNA-seq: Unbiased sampling, Fluidigm C1 | Somatosensory cortex, hippocampal CA1 | **Species/strain:** CD1 mice<br>**Age:** P21–P31<br>**Sex:** both sexes | **Total:** 3005<br>**Astrocytes:** 210 (7%) | **2 subclusters:**<br>- Type 1 (Gfap); glia limitans<br>- Type 2 (Mfge8); uniformly distributed across the cortex |
| Zeisel et al, 2018 | scRNA-seq: FACS neuronal cell type enrichment, 10x Genomics Chromium | 19 brain regions, including: cortex, spinal cord, midbrain, enteric nervous system, dorsal root ganglia | **Species/strain:** CD1, Swiss, Wnt1-CreR26Tomato, Vgat-CretdTomato mice<br>**Age:** P12–P30, 6–8 weeks<br>**Sex:** both sexes | **Total:** 492,949<br>**Astrocytes:** 49,295 (~10%) | **7 subclusters:**<br>- 2 telencephalic clusters (Mfge8, Lhx2)<br>- 2 non-telencephalic clusters (Agt)<br>- olfactory cluster<br>- dorsal midbrain cluster (Myoc)<br>- Bergmann glia |
| Saunders et al, 2018 | scRNA-seq: Drop-seq | 9 brain regions including: frontal/posterior cortex, hippocampus, thalamus, cerebellum, striatum | **Species/strain:** C57Blk6/N, Aldl1l-EGFP, Cx3cr1-GFP mice<br>**Age:** P60-70<br>**Sex:** male | **Total:** 690,207<br>**Astrocytes:** 23,935 | **8 subclusters**<br>(no further characterization) |
| Batiuk et al, 2020 | scRNA-seq: ACSA-2 FACS, Smartseq2 | Cortex, hippocampus | **Species/strain:** C57BL/6J mice<br>**Age:** P56<br>**Sex:** both sexes | **Astrocytes:** 1811 | **5 subclusters:**<br>- 2 hippocampal clusters<br>- 1 cortical cluster<br>- 2 uniformly distributed across cortex and hippocampus |
| Bayraktar et al, 2020 | scRNA-seq: ACSA-2 FACS, Smartseq2<br>Spatial transcriptomics: smFISH | Cortex<br>Coronal brain sections | **Species/strain:** C57BL/6J mice<br>**Age:** P56<br>**Sex:** both sexes<br>**Species/strain:** Swiss Webster mice<br>**Age:** P14, P56<br>**Sex:** not specified | **Astrocytes:** 990 (cortical subset of the Batiuk et al 2020 dataset) | / |
| Ohlig et al, 2021 | scRNA-seq:<br>ACSA-2 MACS, 10x Genomics Chromium<br>Spatial transcriptomics: Visium 10x Genomics (publicly available dataset) | Diencephalon | **Species/strain:** Aldh1l1-eGFP mice<br>**Age:** P90<br>**Sex:** both sexes | **Astrocytes:** 21,503 | **7 subclusters:**<br>- 1 diencephalic restricted cluster<br>- 3 clusters enriched in thalamus and cortex<br>- 1 cluster enriched in lateral diencephalon and hippocampus<br>- 2 clusters less defined |
| Siletti et al, 2023 | snRNA-seq: NeuN FACS, 10x Genomics Chromium | Forebrain (cerebral cortex, hippocampus, cerebral nuclei, hypothalamus, thalamus), midbrain, hindbrain (pons, medulla, cerebellum) | **Species:** human postmortem tissue<br>**Age:** 18–68 years<br>**Sex:** both sexes | **Total:** 3,369,219<br>**Astrocytes:** 155,025 | **2 subclusters:**<br>- telencephalic cluster<br>- non-telencephalic cluster<br>- further distinguished based on Gfap expression |
| Bocchi et al, 2025 | scRNA-seq: 10x Genomics Chromium<br>Spatial transcriptomics: Visium 10x Genomics | Cortical GM, corpus callosum/cerebellum WM, subependymal zone<br>Sagittal brain section | **Species/strain:** C57BL/6J mice<br>**Age:** P60-P90<br>**Sex:** male<br>**Species/strain:** C57BL/6J mice<br>**Age:** P60<br>**Sex:** male | **Total:** 66,455<br>**Astrocytes:** 4541 | **8 subclusters:**<br>- 4 GM clusters<br>- 2 WM clusters from corpus callosum<br>- 2 WM clusters from cerebellum |

This review summarizes research characterizing transcriptomic astrocyte heterogeneity, and aims to critically evaluate the strengths and limitations of various sequencing and analytical approaches, with a call to better align these methodologies to validate—or challenge—transcriptional findings. Further, this review extends the discussion about the astrocyte multistate concept (Escartin et al, 2021), highlighting the importance of multi-omic and longitudinal analyses.

# Insights into astrocyte diversity from transcriptome analysis

## Exploring homeostatic astrocyte heterogeneity in the sc/snRNA-seq era

As mentioned above, astrocyte heterogeneity has been demonstrated at the molecular level showing a broad spectrum of molecular and regional diversity in astrocytes, changing the idea that they represent a uniform cell type (Table 1). One of the first single-cell studies using murine brain tissue analyzed the somatosensory cortex and hippocampal CA1 region, capturing all cell types but retrieving only a relatively small number of astrocytes. This limitation led to the identification of only two astrocyte clusters: one representing astrocytes from cortical layer 1 and the other uniformly distributed across the cortex (Zeisel et al, 2015). In 2018, a comprehensive study analyzing the cellular diversity of the mouse nervous system utilized scRNA-seq data from 19 different regions generated with the 10x Genomics platform (Zeisel et al, 2018). This study identified seven molecularly distinct astrocyte clusters, each exhibiting a clear region-specific distribution. Molecularly distinct astrocytes were identified in the olfactory bulb, cerebellum, telencephalon, midbrain, and other non-telencephalic regions. The genes Agt (Angiotensinogen) and Mfge8 were identified as the primary markers distinguishing astrocytes in the telencephalon from those in the diencephalon. Interestingly, astrocytes from the telencephalic and non-telencephalic regions were the only ones to separate into two distinct clusters, with differences observed, for example, in glial fibrillary acidic protein (Gfap) expression. The authors suggest that the Gfap-high-expressing cluster in the telencephalon may correspond to fibrous astrocytes in the WM and at the glia limitans beneath the pial surface. At the same time, Saunders and colleagues published a study analyzing nine murine brain regions using the Drop-seq method to explore shared and region-specific patterns in cellular composition and gene expression identifying eight different astrocyte clusters, which were not further characterized (Saunders et al, 2018).

### Refining astrocyte diversity through enrichment-based profiling
Early single-cell studies primarily described different molecular clusters of astrocytes. To gain deeper insights into astrocyte heterogeneity, subsequent sc/snRNA-seq studies focused on specific brain regions or employed techniques designed to enrich for astrocytes. Batiuk and colleagues utilized the ACSA-2 antibody to enrich astrocytes from two distinct murine forebrain regions (cortex and hippocampus) of adult mice, identifying five distinct astrocyte clusters (Batiuk et al, 2020). Within these astrocyte clusters, they identified subclusters that were specific to either the cortex (1 cluster) or hippocampus (2 clusters), as well as two others that shared gene expression profiles and spatial distribution across both regions. The same approach was applied to analyze the heterogeneity of diencephalic astrocytes, leading to the identification of seven distinct clusters (Ohlig et al, 2021). Many astrocytic functions, such as ion regulation, sodium transport, and fatty acid or glutamate metabolism, were found to be enriched in distinct clusters. This suggests that these functions may be either transiently distributed among astrocytes in different transcriptional states or stably associated with specific astrocyte subtypes. Furthermore, the authors identified clusters with highly specific spatial localization to a single brain region (diencephalon) and others with broader distribution throughout the forebrain. Thus, only some astrocytes differ profoundly between regions, while others share some pan-astrocyte tasks, such as ion homeostasis (Ohlig et al, 2021). This discovery of shared expression profiles between brain regions, highlighted in these two studies, emerged from the analysis of a larger number of astrocytes per brain region (Ohlig et al, 2021; Batiuk et al, 2020). Moreover, the identification of 7 clusters of diencephalic astrocytes underscores the importance of collecting a large number of astrocytes from a single region, to achieve the resolution necessary for detecting subtle differences in gene expression. Importantly, novel tools to analyze scRNA-seq data, such as velocity analysis based on differences between spliced and unspliced transcripts, led to the discovery of a new function of astrocytes in the diencephalon, namely adult astrogenesis. This was confirmed by incorporation of the DNA-base analog 5-EdU in adult astrocytes and genetic fate mapping demonstrating 2–3 cell clones of astrocytes very close to each other (Ohlig et al, 2021). Finally, the authors showed that adult astrogenesis was Smad4-dependent. This shows how transcriptome analysis inspired the discovery of a novel astrocyte hallmark and function. Notably, however, expression of proliferation genes was not limited to a specific cluster of astrocytes, but spread through all astrocyte clusters in the diencephalon, suggesting that it is a widespread characteristic of astrocytes in this region.

### Astrocyte subtypes in focus: gray versus white matter
A key question in the field of astrocyte heterogeneity is the molecular distinction between GM and WM astrocytes. Furthermore, would WM astrocytes differ between brain regions as GM astrocytes do? Answering this question has proven challenging, as isolating cells from WM brain tissue is complicated by its high myelin content. A recent study developed a new protocol to isolate all cell types from the WM of the corpus callosum (CC) and cerebellum by using a mild dissociation and hence an unbiased approach, avoiding any selection. This was important, primarily because little was known about WM astrocyte markers, and secondly to ensure no subtypes were overlooked (Bocchi et al, 2025). Combining scRNA-seq analysis with spatial transcriptomics of astrocytes from the cerebral cortex GM and WM identified four clusters of GM and two clusters of WM astrocytes. Consistent with previous data (Batiuk et al, 2020; Bayraktar et al, 2020), GM clusters showed layer-specific localization. As in the diencephalon, some astrocyte clusters in this study showed more widespread patterns of gene expression as determined by spatial transcriptomics overlay, suggesting shared molecular characteristics of some astrocyte subsets between different brain regions. WM astrocytes exhibited enriched Gene Ontology (GO) terms related to glycogen metabolism, glycogen breakdown, and the regulation of amide metabolic processes. In addition, genes associated with cholesterol metabolism and cytoskeleton regulation were differentially

expressed between GM and WM. Interestingly, this study also identified a cluster of WM astrocytes in the CC capable of cell division and hence ongoing astrogenesis, a finding that was validated using various methods, including live in vivo imaging. Thus, adult astrogenesis in the intact brain was discovered for astrocytes with distinct molecular identities, i.e., clustered in scRNA-seq data analysis as in the CC WM, or more widespread, as found in the diencephalon. Notably, CC WM astrocytes divide much faster than diencephalic astrocytes with the later resembling more the slow-dividing oligodendrocyte progenitors in the adult brain (Simon et al, 2011). Importantly, this and another study (Seeker et al, 2023) identified subsets of WM astrocytes with region-specific differences. CC WM astrocytes differed significantly from those in the cerebellum, where astrocyte gene expression fell into a cluster similar to CC WM astrocytes and one very different. These findings indicate the existence of region-specific molecular signatures of WM astrocytes in the brain, each potentially serving unique functional roles (Seeker et al, 2023; Bocchi et al, 2025).

Notably, the above-described combination of scRNA-seq data with spatial transcriptomics or multiplexed single-molecule fluorescence in situ hybridization (smFISH) facilitates the discovery of new insights into astrocyte heterogeneity. This has been shown in (Bayraktar et al, 2020) with astrocytes grouped into three spatial bins—superficial, mid, and deep laminae—with genes such as Chrld1, Eogt, Spry1, Paqr6, and Il33 showing distinct layer-specific expression patterns, as well as in (Bocchi et al, 2025) with spatial mapping of scRNA-seq data revealing three GM astrocyte clusters associated with layer 1, the upper layers, and the deep layers, respectively. Altogether, the integration of spatial datasets revealed both layer-independent and layer-dependent heterogeneity, as well as differences across functionally distinct cortical areas (Bocchi et al, 2025; Bayraktar et al, 2020).

### Astrocyte heterogeneity in the human brain

Similar to the mouse brain, human astrocytes can primarily be classified into two major clusters based on snRNA-seq data obtained from the entire brain: telencephalic and non-telencephalic astrocytes. These clusters can be further distinguished by populations with high or low GFAP expression, along with additional clusters, e.g., for striatal astrocytes (Zeisel et al, 2018; Siletti et al, 2023). However, so far the morphological subtypes of astrocytes, e.g., human cortical intralaminar and varicose projection astrocytes, that display a very distinct morphological phenotype (Oberheim et al, 2009), could not be aligned with clusters based on transcription. To achieve this, patch-seq technology may be better suited as discussed below. As previously noted, Seeker et al demonstrated pronounced astrocyte heterogeneity in human white matter, with distinct subsets exhibiting region-specific molecular signatures (Seeker et al, 2023).

However, a detailed molecular characterization of human astrocyte heterogeneity across different brain regions is still lacking, despite various snRNA-seq studies employing cluster analyses and examining differentially expressed genes (DEGs). As these studies are highly influenced by sample quality, number of cells obtained and various analysis parameters affecting resolution, more detailed analyses of human astrocyte subtypes and states in the healthy brain are needed (Colonna et al, 2024).

### Pan-astrocyte functions and the division of labor among subtypes

Another relevant question for understanding astrocyte heterogeneity is how these subtypes resemble one another and to what extent they share common pan-astrocytic functions. For example, while GM astrocytes contribute to BBB maintenance, this function is less evident for WM astrocytes (Hösli et al, 2022). Similarly, the degree of astrocytic coverage at synapses seems to vary across different brain regions (Farhy-Tselnicker and Allen, 2018). Ohlig et al identified distinct astrocyte clusters exhibiting similar enrichment for gene ontology terms associated with ion transport and ion homeostasis, underscoring core astrocytic functions that may be conserved across brain regions (Ohlig et al, 2021).

Using the Ribotag approach, astrocyte bulk RNA-seq revealed around 20% shared gene expression across astrocytes from 13 different brain regions. Half of these genes were related to enzymatic and transporter activity or transcriptional regulation and involved in pathways associated with neurotransmitter homeostasis, cholesterol biosynthesis, and glucose metabolism (Endo et al, 2022). Notably, little is known about one-third of the top genes shared between astrocytes, suggesting that fundamental aspects of core astrocyte functions are still not fully understood (Endo et al, 2022). An alternative strategy to assess shared functions across astrocyte subtypes was employed by Mathys et al, who applied a novel approach (single-cell decorrelated module networks or scdemon) to identify gene expression modules composed of highly correlated genes within a snRNA-seq dataset, thereby uncovering an astrocyte-wide functional program associated with cognitive resilience in Alzheimer's Disease (AD) (Mathys et al, 2024). This gene module analysis offers the advantage of uncovering co-expressed gene networks that define core functional programs, enabling the identification of common versus subtype-specific astrocyte roles.

To explore this theory, we used the scdemon approach to reanalyze the above-mentioned GM and WM scRNA-seq dataset (Bocchi et al, 2025) (Fig. 2), and identified 16 different gene modules, with some showing clear region- and cluster-specific enrichment (GM: module 6; WM: module 2), whilst other modules were more generally expressed (module 4 or 9). GO term analysis [STRING; simona; (Gu, 2024)] of pan-astrocyte module 4 revealed typical astrocyte functions such as signal transduction and metabolic processes. Whereas examination of the region-specific modules revealed enrichment of only certain common astrocytic GO terms (Fig. 2). This could indicate that astrocyte subtypes are not performing all pan-astrocytic functions to the same extent and might be more specialized for some of them, as also suggested in Ohlig et al, 2021. Applying this analytical strategy to additional datasets could provide valuable insight into core astrocyte functions and how they diversify between different subtypes.

## Expanding astrocyte heterogeneity: injury- and disease-induced reactivity

The extensive astrocyte heterogeneity described in homeostatic conditions diversifies even further in response to pathological stimuli. During injury or disease, astrocytes undergo molecular, morphological, and functional changes that are beneficial or detrimental to disease progression and are referred to as "reactive" astrocytes. Whether, how and to what extent astrocytes respond to pathological conditions is depending on many factors such as type -acute versus chronic- of injury/disease, the brain region, timing, age and gender (Bardehle et al, 2013; Escartin et al, 2021; Habib et al, 2020; Lange Canhos et al, 2021; Sirko et al, 2023). To date, it is

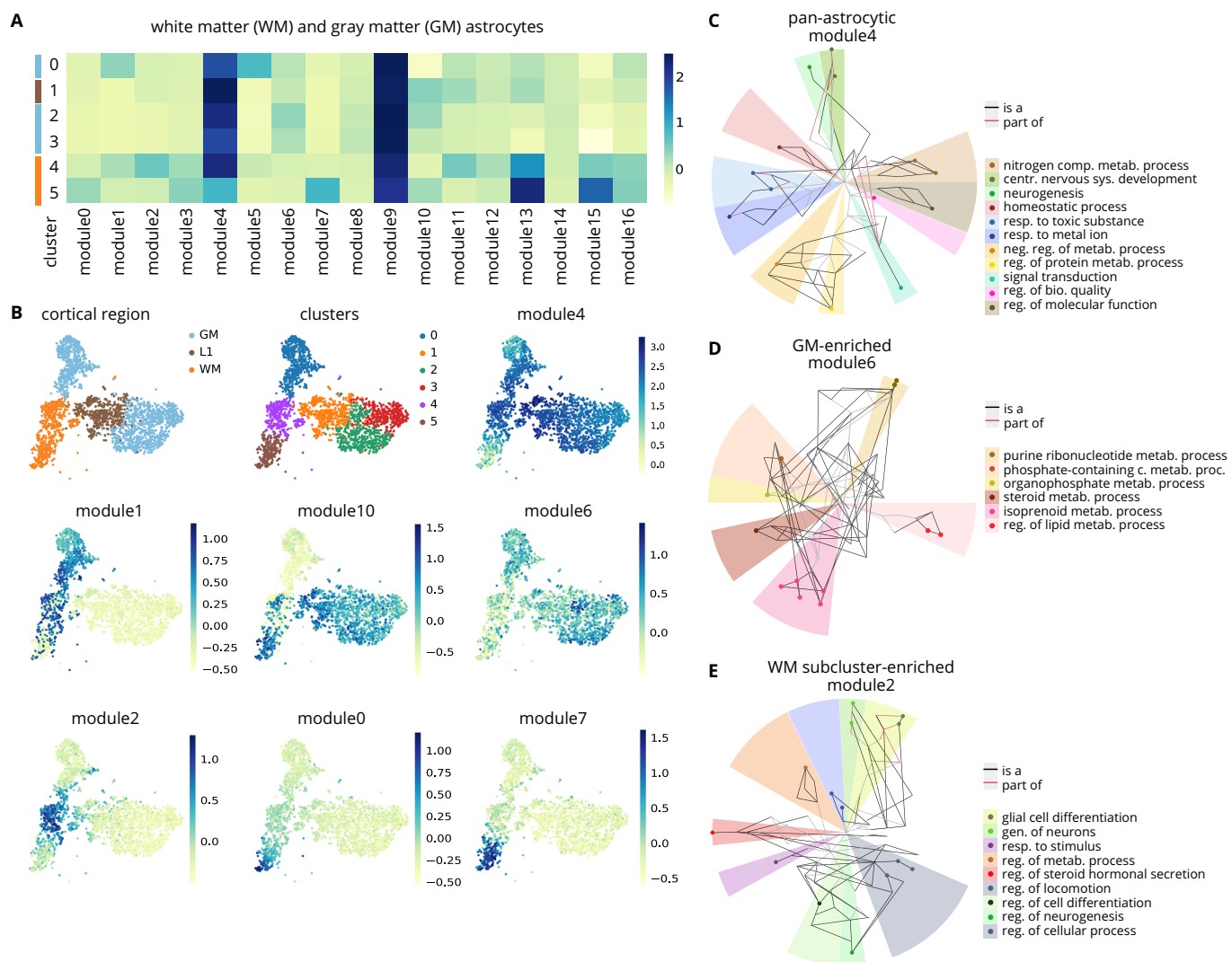

**Figure 2. Using a single-cell decorrelated module networks method to differentiate subtype-specific and pan-astrocyte functions in the data from Bocchi et al, 2025.**

(A) Scdemon identification of 16 gene modules with different expression patterns across GM and WM astrocytes. (B) UMAP plots for astrocytes, colored by brain region, astrocyte subclusters, or gene modules. (C–E) GO term enrichment analysis of pan-astrocyte (C), GM-enriched (D), and WM-cluster enriched (E) gene modules reveals GO functions common across different astrocyte subclusters as well as functions enriched in certain clusters. Colors denote functional groups of GO terms, lines indicate interconnectivity between terms and their hierarchical structure, starting from the middle of the chart, where black lines denote direct parent-child relation of the terms and red lines (C, E) denote terms that are part of a network. GO analysis was done in STRING, selecting the top 15 biological process (BP) terms per module with FDR <0.05. Grouping and visualization of GO terms were done using Simona.

still uncertain whether the reactive response to pathology is a pan-astrocyte feature or if it is limited to a subset of astrocytes and whether a common molecular signature for reactive astrocytes exists across different conditions. The following section provides a brief overview of studies investigating astrocyte heterogeneity in pathological contexts.

### Reactive astrocytes in acute injury conditions

Overall, reactive astrocytes across various pathologies can be classified into two main distinct subclusters; proliferative and non-proliferative astrocytes (Sirko et al, 2023, 2013; Bardehle et al, 2013). This differential response has been attributed to the type of pathology, as mainly conditions where the BBB is disrupted lead to

astrocyte proliferation (Sirko et al, 2023; Sofroniew, 2015). In human pathologies including cerebral cavernoma (CCM), traumatic brain injury (TBI), and stroke, where intracerebral hemorrhage is the common denominator, LGALS3BP was identified as a crucial regulator of this astrocytic response (Sirko et al, 2023). Interestingly, this injury-induced astrocyte proliferation is partly location-dependent. After a traumatic brain and ischemic injury in mouse cerebral cortex, proliferating reactive astrocytes are primarily found with their somata positioned at blood vessels, the juxtavascular astrocytes introduced above (Sirko et al, 2013; Bardehle et al, 2013). This serves two crucial functions: preventing monocyte infiltration and facilitating the restoration of the BBB (Bush et al, 1999; Frik et al, 2018). Following injury, the

electrophysiological properties of juxtavascular astrocytes differ significantly from those of non-juxtavascular astrocytes. This is accompanied by a notable downregulation of K$_{ir}$4.1, a key astrocytic ion channel, predominantly in juxtavascular astrocytes (Götz et al, 2021). These results suggest that in the cerebral cortex, astrocyte subtypes exist with a different predisposition for a reactive response according to their specific position.

Besides proliferation, another feature of some reactive astrocyte states and even of other glial cell states is the activation of innate immune processes after traumatic brain injury in a time-dependent manner (Han et al, 2021; Koupourtidou et al, 2024). A combinatorial approach of single-cell and spatial transcriptomics after stab wound injury in mouse cortex, revealed injury-induced reactive astrocyte states characterized by angiogenesis and immune system processes. Moreover, the astrocytic response to this injury was shown to be heterogeneous with different clusters accumulating at the injury site and displaying different transcriptomic responses (Koupourtidou et al, 2024). Upregulation of genes associated with innate immune processes can also be observed in reactive astrocyte states following stroke models and LPS induction, whereas milder injury models do not trigger such upregulation (Koupourtidou et al, 2024; Arneson et al, 2022; Hasel et al, 2021; Sirko et al, 2023; Zheng et al, 2022).

### Astrocyte heterogeneity in chronic disease

Whereas hemorrhage-driven pathologies elicit localized astrocyte proliferation, snRNA-seq studies in AD reveal a complex and region-specific astrocyte response that reflects the chronic and heterogeneous nature of neurodegeneration (Cain et al, 2023; Green et al, 2024; Grubman et al, 2019; Lau et al, 2020; Mathys et al, 2024; Serrano-Pozo et al, 2024). As mentioned above, a study examining six distinct brain regions from AD patients and controls investigated cellular diversity upon ageing and the responses of various cell types to the disease, detecting region-specific astrocyte clusters that were also present in other datasets, and discovering 32 gene modules in astrocytes (Mathys et al, 2024). The authors identified a module of reactive astrocytes in response to plaque burden and found that astrocytes exhibited a higher number of plaque-associated DEGs—many of which were linked to metallostasis, compared to other cell types. Notably, astrocytes were the only cell type expressing genes associated with cognitive resilience. These promote antioxidant functions, suggesting a unique contribution of astrocytes to antioxidant defense mechanisms (Mathys et al, 2024). On the other hand, in an AD mouse model, a novel astrocyte subtype termed disease-associated astrocyte (DAA), characterized by high Gfap expression, was discovered. These DAA's accumulate during disease progression and show a unique expression of genes involved in endocytosis, complement cascade and ageing (Habib et al, 2020). These studies underscore that astrocyte responses are not uniform, even within the same disease context.

Thus, the reactive response of astrocytes in injury/disease conditions is highly heterogeneous and depending on many factors. Unraveling astrocyte diversity in pathological conditions is critical for the development of novel therapeutic strategies. As not all astrocytes seem to respond in a similar fashion, this suggests a potential difference in the underlying vulnerability of astrocyte subtypes to environmental changes. Obtaining a better understanding the heterogeneity of reactive astrocytes will be aided by a comprehensive picture of astrocyte heterogeneity under healthy conditions, including the influence of gender and the respective changes that occur with aging. To address this, it is essential to understand the underlying reasons for the limited consensus among astrocyte heterogeneity studies, and to identify strategies for improving consistency.

# Key considerations for the interpretation of astrocyte transcriptome data

The recent surge in astrocyte transcriptomic studies has provided us with a deeper understanding of the complexity of molecular astrocyte heterogeneity. However, it has also left us with many hurdles to align the different findings and to comprehend the dynamic nature of the many astrocyte subtypes. This necessitates stepping back to reevaluate experimental set-ups and fundamental aspects of astrocyte physiology, ensuring accurate interpretation of this extensive body of information.

## Experimental parameters and strategies: what information are we extracting?

### Single cell vs single nuclei

Discrepancy in the observed astrocyte subtypes is largely attributed to differences in experimental parameters and strategies. Evidently, inconsistencies appear between studies that analyze different cellular compartments. ScRNA analysis captures and analyzes RNA content largely from the cell soma, while snRNA studies will only provide information about nuclear transcripts. A comparative analysis between the two approaches showed that in pyramidal neurons, the nuclear portion of total cellular mRNA varies from 20 to 50%. This led to a lower number of transcripts detected in snRNA (~7000/nuclei) compared to scRNA analysis (~11,000/cell) (Bakken et al, 2018). Including intronic sequences in the analysis was shown to be imperative to distinguish similar neuronal subtypes in the single-nuclei dataset (Bakken et al, 2018). Retrieving similar astrocyte subtypes with snRNA analysis as with scRNA analysis could be even more complicated as astrocytes have smaller nuclei than neurons, which correlates with RNA transcript levels (Mohammadi et al, 2023; Webster et al, 2009).

Both approaches have distinct advantages and limitations, when interrogating astrocyte diversity. Most studies investigating astrocyte heterogeneity in the murine brain have implemented a single-cell approach as it provides a higher amount of transcripts/cell and has a cell capture rate biased towards glial cells. However, single-cell methods often induce artificial transcriptomic perturbations as cells are vulnerable to the dissociation protocols (Marsh et al, 2022; Mattei et al, 2020). On the other hand, single-nuclei isolation causes less cellular stress but is highly biased towards neuronal cell capture, potentially due to their larger size. As single nuclei can be isolated not only from fresh tissue but also from frozen and fixed tissue, most human transcriptome studies investigating astrocyte heterogeneity have implemented this method making cross-species comparison very challenging.

### Protocol improvements: sequencing workflows

Building on the pioneering single-cell study by the Linnarsson group (Zeisel et al, 2015), numerous efforts were made over the past

decade to refine sc/snRNA-seq workflows to enhance both the quantity and quality of captured cells. First, scRNA-seq methods differ in how they tag transcripts and generate sequencing libraries. Low-throughput plate-based methods sort a cell into a well of a multi-well plate and involve full-length cDNA amplification to obtain high-quality deep-sequencing data from individual cells, allowing detection of rare transcripts and isoforms. High-throughput bead-based methods, on the other hand, are more efficient for analyzing a large number of cells but at the expense of sequencing depth. These methods distribute cell suspensions into droplets or wells that contain reagents and barcoded beads. The generated single droplets/wells will encompass one bead labeled with oligonucleotides for capturing the transcripts present in each cell. A systematic comparison between the different available methods suggests that for low-throughput methods, Smartseq2 would be the method of choice, whereas for the high-throughput methods, 10X Chromium was the top performer (Ding et al, 2020). Many astrocyte heterogeneity studies have implemented high-throughput methods to be able to analyze a large number of astrocytes to retrieve subtypes. However, these could miss important information, such as splice isoforms. For example, astrocyte Gfap isoform transcript levels differ between brain regions, developmental stages and disease states (Kamphuis et al, 2012). More recently, an enhanced single-cell long-read method (ScISOr-Seq2) was developed allowing full-length isoform analysis across thousands of individual cells, effectively combining the high sensitivity of low-throughput methods with the cellular coverage of high-throughput platforms. Using this strategy, astrocytes were found to show complex isoform variability patterns along regions, ages, and subtypes (Joglekar et al, 2024). Thalamic and cerebellar astrocytes were shown to have a high degree of specialized isoform expression compared to other regions. Given that the cerebellum harbors a morphologically and functionally distinct astrocyte subtype, the Bergmann glia, these findings hint towards a role for alternative splicing in specialized astrocyte subtypes (Joglekar et al, 2024). By providing more comprehensive transcriptomic information, this approach holds great potential to advance our understanding of astrocyte heterogeneity under both physiological and pathological conditions.

### Protocol improvements: dissociation workflows

Besides the development of new methods for cell capture and sequencing, many improvements to the dissociation protocol have been made to enhance the amount and the quality of the captured cells. As previously mentioned, tissue dissociation can introduce artificial transcriptomic perturbations, which may hinder the accurate detection of baseline transcriptional profiles as well as condition-induced acute transcriptome changes. In particular, a dissociation-triggered upregulation of immediate early genes (IEGs) is observed, causing an artificial activation signature in isolated cells. By introducing a general transcription inhibitor, actinomycin D, during the isolation process, a more faithful detection of transcriptomic changes can be achieved (Liu et al, 2021; Wu et al, 2017; Safaiyan et al, 2021). In addition, dissociation of brain tissue requires harsh conditions that evidently leads to a certain degree of cell death. Dead cells can lyse easily, resulting in the release of ambient RNA which potentially leads to background noise and compromises single-cell data quality. Removal of dead cells from single-cell suspensions can significantly improve the

performance of 10x Genomics experiments. By introducing a debris/dead cell removal step, samples showed increased cleanliness, accuracy in target cell count, library complexity, and a decreased mitochondrial contamination (Bocchi et al, 2025).

To collect a sufficient number of astrocytes, enrichment strategies such as fluorescent or magnetic activated cell sorting (FACS/MACS) can be used. Enriching astrocyte numbers has been successfully achieved by FAC-sorting using astrocyte-specific reporter mouse lines such as the Aldh1l1-eGFP mouse line (Hasel et al, 2021; Kim et al, 2023). On the other hand, MACS isolation of astrocytes can be achieved using the ACSA-2 magnetic beads (Miltenyi Biotec) (Ohlig et al, 2021; Scott et al, 2024). The ACSA-2 epitope was identified as Atp1b2, considered to show stable astrocyte expression in multiple models of CNS injury and disease (Batiuk et al, 2017). However, more recently it was shown that ACSA-2 MACS isolation of astrocytes leads to a significant contamination of ependymal cells (Ohlig et al, 2021). Direct comparison of the two techniques revealed that MACS isolation results in a lower percentage of cell loss (7–9%) compared to FACS (70%) and can process samples 4–6 times faster and allows parallel processing of samples. On the other hand, FACS has been shown to produce samples with higher purity and supports multi-marker analysis (Pan and Wan, 2020; Sutermaster and Darling, 2019). Even though both protocols produce samples with high cell viability, they significantly prolong the isolation process resulting in a certain degree of cellular stress and artificial transcriptome changes. Moreover, it is unclear if the markers used for astrocyte enrichment might be differently expressed between distinct subtypes and thus lead to the preferential targeting of certain subtypes. An unbiased sampling approach is therefore the favored strategy to avoid selecting or missing astrocyte subsets and to maximize cell survival and quality. Conversely, when the objective is to isolate a specific subtype, an adapted enrichment strategy is required. For example, McCarty et al developed a Mlc1-eGFP transgenic mouse strain to specifically label astrocytes in contact with blood vessels, enabling the investigation of their role in regulating vascular function in health and disease (Yosef et al, 2020; Morales et al, 2022; Toutounchian and McCarty, 2017).

### Bioinformatics analysis

In addition to the diversity in the technical and practical aspects of sc/snRNA analysis, a wide range of approaches is also available for bioinformatic data analysis. Many different streamlined analysis tools exist, such as Seurat (Hao et al, 2024) or Scanpy (Wolf et al, 2018), that are accompanied with a user-friendly tutorial, making sc/snRNA data analysis accessible to everyone. One drawback of these programs is that many parameters are user-defined, which can significantly impact the degree of clustering. To learn effective strategies for avoiding this type of bias in analysis, we recommend consulting the recent review by Colonna et al (Colonna et al, 2024). Apart from the classical analysis pipelines, recent advances in the field led to the development of several cell–cell communication algorithms. These methods can give insights into the interactions between astrocytes and their surrounding cells and thereby provide information about the impact of the respective niche. For instance, Mathys et al recently reported shared cell–cell communication across multiple brain regions, but region-specific neuronal signaling in the thalamus (Mathys et al, 2024). For an overview of current cell–cell communication algorithms for different omic layers, we

recommend the review by Armingol et al (Armingol et al, 2021). To facilitate more meaningful comparisons across datasets, bioinformatic analyses should also be standardized.

## Bypassing the caveats of tissue dissociation

### Spatial transcriptomics

Despite numerous efforts to optimize quality and maximize output in sc/snRNA-seq, tissue dissociation inevitably introduces bias into the transcriptomic readout. Of note are techniques that bypass this dissociation process and allow investigation of cellular heterogeneity in situ. Spatial transcriptomics enables a quantitative readout of gene expression mapped to specific locations in a tissue section. While sc/snRNA analysis aids us in identifying individual cellular puzzle pieces, spatial transcriptomics can reveal their spatial relationships and how they fit together within the tissue. Considering that astrocytes have a positional identity, studying them in their natural cellular surrounding holds great promise to further unravel their heterogeneity.

Spatially resolved transcriptomics can be broadly divided into sequencing-based and imaging-based technologies that differ in capture area, sensitivity, number of genes profiled, and resolution. Sequencing-based spatial methods are high-throughput and can map the whole transcriptome to the tissue, but often lack sensitivity and single-cell resolution (Bressan et al, 2023; Tian et al, 2023; Valihrach et al, 2024). Imaging-based approaches, on the other hand, reach subcellular resolution while they are limited by the number of measured transcripts as well as the capture area size (Valihrach et al, 2024; Bressan et al, 2023; Tian et al, 2023). These limitations complicate the identification of astrocyte subtypes, which are frequently defined by subtle transcriptomic changes. Therefore, spatial transcriptomics is more commonly used as a complementary approach to sc/snRNA analysis, enabling the mapping of identified subtypes back to their spatial context within the tissue. For example, this complementary strategy revealed differences in the spatial distribution of GM and WM astrocyte clusters, with some showing a more widespread, and others a more localized distribution pattern (Bocchi et al, 2025). In the context of reactive astrocyte heterogeneity, this approach offers insights into the relationship between subclusters and their proximity to the injury site (Koupourtidou et al, 2024). Importantly, many of the spatial transcriptomics technologies can be combined with measurement of epigenome, proteome or metabolome (Vandereyken et al, 2023). Such a multi-omics analysis could be highly valuable in further elucidating astrocyte diversity (see below).

### Patch-sequencing

Another strategy for studying astrocyte subtypes in situ is patch-seq, which enables simultaneous measurement of whole-cell electrophysiological recordings, scRNA transcriptome as well as morphological parameters. Briefly, a Giga-Ohm seal is established between the pipette and the cell as in patch-clamp electrophysiology and can then be used to record (or not) and later extract the cytoplasmic content via the recording pipette (Natarajan et al, 2024). The extracted material is further subjected to scRNA-seq to obtain the cell's gene expression profile. Lastly, a labeling strategy or loading dye is used to visualize detailed cellular morphology (Lipovsek et al, 2021; Shao et al, 2023; Cadwell et al, 2016). This multimodal approach has generally been used to unravel neuronal

heterogeneity across brain regions and species (Shao et al, 2023). Although astrocytes do not generate action potentials, they are electrically dynamic cells with a high degree of electrophysiological heterogeneity (McNeill et al, 2021). Whole-cell patch recordings of juxtavascular and non-juxtavascular astrocytes after injury revealed differential electrophysiological properties potentially related to the increased proliferation of the juxtavascular subtype (Götz et al, 2021). Interestingly, patch-seq of morphologically distinct astrocyte subtypes revealed molecular differences between ventricular zone-derived cortical plate astrocytes and outer subventricular zone-derived WM astrocytes (Allen et al, 2022). As this method has also been successfully used to characterize astrocytes targeted via a viral vector approach for labeling and/or genetic modification, it could in addition be used to analyze astrocyte subtypes with a distinct morphology, e.g., the human astrocytes using acute slices from human brain tissue. The need for prior labeling is however also a drawback of this technology as many subtypes are characterized by subtle differences in gene expression and lack specific markers. However, viral vector labeling of astrocytes using e.g., a GFAP-driven promoter may be used in adult human brain slices, to explore astrocytes with different morphologies. Indeed, it is still an open question to which extent morphological differences between astrocytes are reflected by molecular distinctions.

In summary, many different approaches exist to investigate astrocyte heterogeneity that extract different information that makes comparison between datasets extremely challenging. On top of that, many protocols have been optimized over time to maximize cell amount and cell quality, making correlations with earlier studies tricky. The preferred strategy for investigating astrocyte diversity will primarily depend on the scientific question. Overall, shortening and optimizing workflow protocols will produce better quality datasets and using an unbiased cell capture approach will avoid potential subtype targeting.

Regardless of the analysis strategy used, it is essential to validate the identified subtypes beyond their transcriptomic signatures. Ideally, an astrocyte subtype should have distinct morphological, molecular, and functional features. While multi-level validation of identified subtypes would be ideal, it has proven to be technically challenging. A comprehensive overview of all possible validation experiments and their challenges/limitations can be found in Colonna et al (Colonna et al, 2024).

## Astrocyte biology: what information are we missing?

### Local translation

A key point to consider is that the majority of these studies define distinct subtypes solely based on RNA content differences in the cell soma or nucleus, overlooking critical information from other subcellular compartments of astrocytes. A unique feature of astrocytes is their branched morphology that allows them to contact and regulate blood vessels via their endfeet and synapses/dendrites via their peripheral processes. These protrusions are highly specialized structures, containing a variety of transporters, channels and neuroactive substances equipping them to sense their environment and coordinate local neuronal activity (Boulay et al, 2017; Murphy-Royal et al, 2017). Moreover, astrocyte processes exhibit localized microdomain calcium transients that correlate with changes in metabolic support and neurovascular coupling (Agarwal et al, 2017; Gau et al, 2024; Otsu et al, 2015).

Interestingly, Sakers et al discovered that these fine distant astrocyte processes contain a local translation machinery allowing astrocytes to locally translate proteins capable of affecting surrounding synapses (Sakers et al, 2017). By using a translating ribosome affinity purification (TRAP) strategy, they identified that transcripts localized in these distant processes show an enrichment for certain biological functions such as fatty acid synthesis, GABA/Glutamate metabolism, and synapse refinement. Moreover, they also locally translate cytoskeletal proteins possibly involved in morphological remodeling of their processes (Sakers et al, 2017). A similar observation was made for astrocyte perivascular processes, where a select subset of mRNA's is locally translated that primarily encodes for secreted and membrane proteins which are involved in vascular homeostasis (Boulay et al, 2017).

The use of astrocyte-ribotag mouse models allowed pulldown and analysis of ribosome-associated mRNAs from all cell compartments and revealed transcriptome differences between astrocytes from different brain regions and across development (Boisvert et al, 2018). In mouse hippocampus, it was shown that ribosome-bound mRNAs in the astrocyte processes, compared with the ones present in the whole astrocyte, are enriched in mRNAs that encode proteins involved in iron homeostasis, translation, cell cycle and cytoskeleton and the composition of these mRNAs are subject to change in memory and learning conditions (Mazaré et al, 2020). In addition, local translation in processes is dynamically and rapidly regulated by neuronal activity and affects astrocyte contributions to tripartite synapses (Sapkota et al, 2022).

These studies highlight that crucial information is contained in astrocyte processes and endfeet that is essentially overlooked in sc/snRNA analyses due to dissociation-induced loss of processes. As many astrocyte subtypes show distinct morphologies with different numbers and complexities of protrusions that results in different levels of synaptic coverage (Chai et al, 2017; Genoud et al, 2006; Herde et al, 2020; Lanjakornsiripan et al, 2018) and perivascular coupling (Hösli et al, 2022), this suggests that different subtypes have different degrees of local translation. To what extent local translation contributes to astrocyte heterogeneity is still largely unexplored. Spatial transcriptomics, where transcriptome analysis is conducted on intact tissue, bypasses the issue of dissociation-induced loss of processes but lacks the resolution to investigate astrocyte protrusions (Mohammadi et al, 2023; Williams et al, 2022). Interestingly, Zeng et al recently developed ribosome-bound mRNA mapping (RIBOmap), a highly multiplexed method that allows spatial characterization of protein translation at the single-cell and subcellular level. They reveal that RIBOmap is capable of distinguishing between transcripts present in the processes or the soma of both neurons and astrocytes (Zeng et al, 2023). It would be interesting to map previous sc/snRNA data to this RIBOmap to reveal if additional transcripts are detected at the level of the processes of the different astrocyte subtypes.

### Discrepancy between RNA and protein levels

A major concern in astrocyte heterogeneity research is that many discovered subtypes can only be identified based on RNA transcripts and largely lack validation at the protein level. On one hand, this is due to a lack of antibodies for the subtype-specific marker, and on the other hand, this is related to a discrepancy between RNA and protein levels. This discrepancy is well-documented and attributed to many processes such as post-

transcriptional regulation, translation control, and protein stability and degradation (Buccitelli and Selbach, 2020). Interestingly, neural stem cells show a particularly high correlation between mRNAs and proteins, suggesting that they contribute to generating their own niche (Kjell et al, 2020).

For astrocytes, the correlation between protein abundance and RNA expression appears to be weak (Soto et al, 2023). This direct comparison between proteome and transcriptome also revealed previously unknown molecules and pathways in astrocyte sub-compartment proteomes, emphasizing the relevance of exploring protein levels in parallel to gene expression data (Soto et al, 2023). Proteomic analysis of astrocytes could be instrumental to further delineate the different subtypes, but is still scarce due to the ongoing development and optimization of single-cell proteomics. A comparison of the proteome between different CNS cell types revealed that only a tenth of the cellular proteome detected is actually cell-specific and that these proteins are mostly cell surface proteins (Sharma et al, 2015). More recently, an in vivo cell-specific biotinylation approach, using the biotin ligase TurboID, investigated proteomic profiles for astrocytes in different brain regions, showing differences between cortex and hippocampus, pons, cerebellum, and spinal cord (Rayaprolu et al, 2022). Proteomic analysis of CNS cell types has been almost exclusively performed on the bulk/population level. However, recent advancements in the field of single-cell proteomics, hold great promise to further unravel cellular heterogeneity (Bennett et al, 2023).

### Functional validation of molecularly identified astrocyte subtypes

Another limitation of sc/snRNA-seq astrocyte heterogeneity studies is the insufficient functional validation, which is critical for confirming the physiological relevance of the molecularly identified astrocyte subtypes. Functional characteristics of astrocyte subtypes have primarily been described between distinct brain regions. Chai et al elegantly demonstrated that in the adult mouse brain, transcriptionally and proteomically distinct hippocampal and striatal astrocyte subtypes also exhibit divergent electrophysiological characteristics, calcium signaling dynamics, and spatial relationships to synapses (Chai et al, 2017). By using imaging sensors for redox state or ATP generation to investigate metabolic specialization of astrocyte subtypes, differences in basal metabolism of WM and GM astrocytes were revealed (Köhler et al, 2023). Also, in vitro models have been implemented to demonstrate functional heterogeneity of regional astrocytes. Human induced pluripotent stem cells (iPSCs), patterned to dorsal and ventral forebrain or spinal cord progenitors prior to astrocyte differentiation, revealed not only distinct regional transcriptomic profiles but also differential physiological properties such as $Ca^{2+}$ signaling and effects on neurite growth and blood-brain barrier formation (Bradley et al, 2019). Such an elaborate functional investigation is often not possible for the identified subtypes within a certain region, as immunological or genetic labeling cannot easily be achieved. Nonetheless, some studies have succeeded in obtaining functional insights, corresponding to transcriptomic profiles, for a subset, but not all, of the identified astrocyte subtypes. Batiuk et al revealed differential $Ca^{2+}$ signaling across cortical layers and CA1 hippocampus and could correlate this back to their previously identified astrocyte subtypes by using their in situ hybridization (ISH) data (Batiuk et al, 2020). In the hippocampus, an astrocyte subcluster was discovered that selectively expressed synaptic-like glutamate-

release machinery. By using GluSnFR (glutamate-sensing fluorescent reporter)-based glutamate imaging a corresponding astrocyte subtype could be revealed that responds to astrocyte-selective stimulations with subsecond glutamate-release events (de Ceglia et al, 2023). In Bocchi et al a WM astrocyte subtype with a molecular signature indicative of ongoing proliferation was identified, which was validated using 5-EdU incorporation, viral labeling as well as live imaging (Bocchi et al, 2025).

All techniques mentioned above are valuable tools for advancing astrocyte heterogeneity research by allowing for a functional characterization of identified astrocyte subtypes. Another promising strategy to retrieve functional readouts of astrocyte subtypes is CaMPARI, or Calcium-Modulated Photoactivatable Ratiometric Integrator. This calcium-sensitive fluorescent reporter that irreversibly changes color in active cells exposed to light, enables time-locked labeling of cellular activity (Moeyaert et al, 2018). When applied to astrocytes, CaMPARI enables spatially resolved mapping of calcium activity, allowing functional responses to physiological or pathological stimuli to be assessed across brain regions. Coupling this approach with RNAscope or in situ hybridization would permit the visualization of distinct functional profiles among astrocyte subtypes within the same region.

## Astrocyte heterogeneity: current understanding and future directions

As became evident from the above-described transcriptomic approaches, the astrocyte population is a lot more complex and diverse than originally appreciated. However, due to the high variety of experimental approaches used to analyze molecular astrocyte diversity, alignment of the datasets and consensus about the number/identity of astrocyte subtypes has proven challenging. As a result, a clear and unified understanding of astrocyte heterogeneity has yet to be established.

Evidence supports the existence of distinct astrocyte subtypes between brain regions, characterized by morphological, molecular, and functional differences. For example, the distinction between GM and WM astrocytes is now appreciated at multiple levels, with the transcriptomes suggesting mechanisms for their distinct morphologies and functions (Bocchi et al, 2025). On the other hand, transcriptomic differences amongst astrocytes within the same region are more nuanced and therefore even more strenuous to validate. Should we revisit the classification of a subtype and what should rather be considered a substate (Fig. 3)?

As outlined in the preamble, for the classification of subtypes, we can draw from neuronal literature, where subtype classification has proven more straightforward. Neuronal subtypes can differ by morphology, neurotransmitter phenotype, physiological properties, connectivity, and expression of specific markers (Molyneaux et al, 2007; Zeng and Sanes, 2017). Importantly, these hallmarks are rather stable, with few changes occurring in the adult brain. Although many aspects of neuronal subtypes are shared, such as similar morphologies and neurotransmitters of cortical neurons, they are distinguished by their unique and stable projection patterns, which persist throughout life (Di Bella et al, 2024; Lodato et al, 2015).

In this context, a key distinction is that subtypes should exhibit stable characteristics, whereas substates reflect more transient

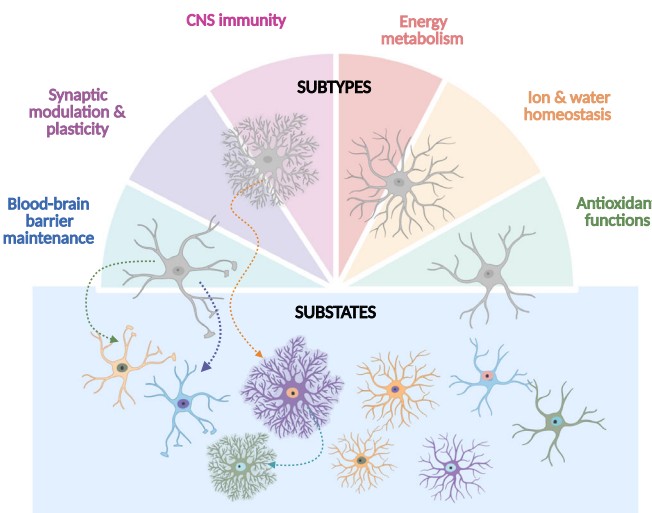

**Figure 3. Graphical representation of the current view of astrocyte diversity.**

Astrocytes are a heterogenous cell type in the CNS, capable of performing a wide range of important tasks (indicated in the figure) to maintain brain homeostasis. Distinct astrocyte subtypes can exhibit unique morphologies and/or functional profiles or demonstrate an enhanced capacity of specific pan-astrocyte functions. Environmental changes or stimuli can trigger transcriptomic alterations, leading to the emergence of a reversible astrocyte substate potentially optimized for specific functional demands. Created in BioRender. Hennes M (2025). https://BioRender.com/55vvjuf.

features. Interestingly, in the case of injury/disease-induced reactive astrocyte subtypes, there is already evidence supporting astrocyte substate transitions. Nearly a decade after the initial proposal of the binary 'A1' neurotoxic and 'A2' neuroprotective astrocyte subtypes induced by injury/disease (Clarke et al, 2018; Liddelow et al, 2017), Zhang et al have now demonstrated—through time-series monitoring combined with multi-omics analyses—that these neurotoxic and neuroprotective phenotypes are in fact substates of the same exact astrocyte and thus not independent subtypes (Zhang et al, 2025). These findings are in line with the multistate concept of astrocytes that was previously introduced for reactive astrocytes capable of adopting multiple states depending on the context (Escartin et al, 2021). Whether astrocyte substates also exist under homeostatic conditions, and how neuronal activity would influence this, is still unresolved. However, as both subtypes and substates are reflected in differential gene expression, their distinction cannot be based solely on transcriptome differences. Ideally, longitudinal studies should determine, how stable or dynamic a gene expression or functional trait may be.

A further important criterion is function. An authentic astrocyte subtype should be defined by their specific functional roles rather than exclusively their transcriptome profiles, as previous efforts to delineate astrocyte heterogeneity have struggled to validate these subtypes at the functional level. Recently, Shainer et al revealed that in zebrafish molecularly similar neurons can be functionally and morphologically diverse (Shainer et al, 2025). They hypothesize that functional and morphological diversity can manifest during differentiation due to restrictions in the local environment, such as the availability of nearby neurons, which might not be reflected in the cell's gene expression levels. This could provide an explanation as to why location-restricted

**Box 1.  Questions to be further pursued in the field of astrocyte heterogeneity**

| Questions to be further pursued | Suggested experimental strategies | References |
|---|---|---|
| How to identify a true astrocyte subtype versus an astrocyte 'substate'? | Longitudinal multi-omic analysis (beyond transcriptome), including a functional characterization | Zhang et al, 2025; Escartin et al, 2021; Shainer et al, 2025; Bandler et al, 2022 |
| How to functionally characterize astrocyte subtypes? | Depending on the molecular findings; Ca²⁺-signaling, electrophysiology, 5-EdU analysis, synaptic coverage, live imaging, CaMPARI | Chai et al, 2017; Bradley et al, 2019; Batiuk et al, 2020; de Ceglia et al, 2023; Bocchi et al, 2025; Moeyaert et al, 2018 |
| What is the origin of astrocyte heterogeneity? | Lineage tracing, TrackerSeq, environmental signals, e.g., from neurons, blood vessels,... | Bandler et al, 2022 |
| To what extent does the local environment define the subtypes? | Transplantation, subtype validation in mouse models with altered environment, e.g., neuronal layers (f.i. Reeler mice) | Bayraktar et al, 2020; Farmer et al, 2016 |
| What are the real pan-astrocyte functions, and is there a division of labor of these across the subtypes? | Transcriptome comparison between clusters, gene module analysis (scdemon) across multiple datasets | Ohlig et al, 2021; Mathys et al, 2024 |

tracing in combination with scRNA-seq, could be an interesting strategy to investigate whether cells labeled in the adult brain shift between clusters in a longitudinal study (Bandler et al, 2022). However, this approach is best applied to fast-dividing cells, such as the cluster observed in the WM, and more difficult to interpret once astrocytes stop dividing. Besides studying astrocytes at different omics levels, one should also consider important information residing in different subcellular compartments, such as the fine distant astrocyte processes. In addition, live imaging approaches would allow observation of astrocyte subtype behavior in their natural environment as well as real-time visualization of potential substate transitions that could be accompanied by changes in Ca²⁺-signaling, blood vessel association or synaptic coverage (Bernardinelli et al, (2014); Bindocci et al, 2017; Mills et al, (2022)).

Finally, it is important to emphasize that despite the many unresolved questions (Box 1) regarding astrocyte diversity, the emerging multi-omic era will be instrumental in unraveling the remaining uncertainties and provide us with groundbreaking insights into astrocyte biology. Given the critical roles astrocytes play in maintaining brain function in both health and disease, a comprehensive understanding of their biology could open the door to novel and effective treatments for neurological disorders.

## Graphics

Biorender was used to create the figures of this review manuscript.

## Peer review information

morphologically distinct astrocyte subtypes, such as the interlaminar and varicose projection astrocytes, cannot be distinguished based on their transcriptome. Another assumption would be that certain cellular characteristics such as morphology are actually regulated at the posttranslational level and/or by low-abundance transcripts that have not yet been detected. Indeed, patch-seq analysis revealed a disconnect between physiological function such as neuronal firing and expression levels of channels or the remnants of astrocyte gene expression (Kempf et al, 2021).

While implementing neuronal frameworks to better grasp astrocyte subtypes can provide valuable insights, they do not convey the full picture. Historically, drawing direct comparisons between astrocytes and neurons has had limited success. The expectation that astrocytes should mirror neuronal electrical activity to be considered functionally relevant led to their prolonged dismissal as passive structural cells. More recently, it was shown that even conserved signaling pathways elicit different functional outcomes as Gi protein-coupled receptor  signaling inhibits neuronal activity but activates astrocytes (Durkee et al, 2019). It is therefore important to recognize that given their distinct biology, astrocytes often require alternative experimental strategies for their investigation.

In light of this, future studies should be advised to implement a longitudinal multi-omics analysis. TrackerSeq, allowing lineage

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

## Acknowledgements

We would like to thank Matteo Puglisi for his contribution to the graphical design and our lab colleagues for their valuable discussions. MG acknowledges funding from the Helmholtz Association, the European Union (NSC Reconstruct project 874758 and advanced ERC NeuroCentro project 885382), and the German Research Foundation (SyNergy project 390857198, TRR274 project 408885537, Immunostroke project 40535880, and Ferroptosis project 461629173).

## Author contributions

**Maroussia Hennes**: Conceptualization; Writing—original draft; Writing—review and editing. **Maria L Richter**: Visualization; Data analysis. **Judith Fischer-Sternjak**: Conceptualization; Writing—original draft; Writing—review and editing. **Magdalena Götz**: Conceptualization; Visualization; Writing—original draft; Writing—review and editing.

## Disclosure and competing interests statement

The authors declare no competing interests.

