## [Peer Review File · EMBO Reports]

Astrocyte diversity and subtypes: Aligning transcriptomics with multimodal perspectives

Maroussia Hennes, Maria Richter, Judith Fischer-Sternjak, and Magdalena Götz

Corresponding author(s): Magdalena Götz (magdalena.goetz@helmholtz-munich.de) , Judith Fischer-Sternjak (judith.fischer@helmholtz-munich.de)

Review Timeline:

Submission Date:	20th Mar 25
Editorial Decision:	2nd May 25
Revision Received:	13th Jun 25
Accepted:	7th Jul 25

Editor: Esther Schnapp

Transaction Report:

Dear Magdalena,

Thank you for the submission of your Review to EMBO reports. We have now received the full set of referee reports that is pasted below.

It is good to see that the referees find the review interesting and timely. However, they do have several comments and suggestions for how the review could be improved, and I think all comments are good and should be addressed, if you agree, of course. If you disagree, we can also discuss the revisions in a video chat, if you like.

I agree with referee 1 that Figure 2 is not sufficiently clear. My understanding is that you re-analyze or re-visualize published data, and this needs to be explained clearly in the review text and the figure legend. While reviews cannot present novel data, I think a re-analysis of published data is probably OK.

Can you please add a Box to the review called "In need of answers". This Box should list open questions in the field, which can be accompanied by suggested experimentation addressing these questions.

As for timing, would it be possible for you to submit the revised version by June 2nd? If you anticipate a problem meeting this deadline, then please just let me know.

Aleksandra Pekowska has not yet submitted her review, so your reviews will not be published back to back.

With best wishes,
Esther

Referee #1:

This is an interesting and timely review on heterogenous astrocyte gene expression and its meaning for distinct astrocyte subtypes versus substates.

There are a number of points that need improvement.

- 1) The text on page 6-7, "In general... " until '...are needed (Colonna2024)" is unclear. The purpose and conclusion of this part is not clear and not adding much. It is describing data of Figure 2 but that text and figure is also not clear:
- 2) Figure 2. This figure should be self-explanatory, with the legend describing: 1) the source of the data, 2) the methods used, and 3) the meaning of lines and symbols. Panels C, D, E: what is the meaning of 'is a', and of 'part of'? Also the red lines of 'part of' are not seen. Mistake in 'SiNgal transduction'. Remove the space in '...using simona ."
- 3) Figure 2. The pan-astrocyte module 4 was described to reveal typical astrocyte functions such as signal transduction and metabolic processes. How significant is this? Also module 6 and 2 (GM and WM) have terms that could be considered pan-astrocyte, including 'steroid hormone' terms. It is not surprising anyway that different modules have different GO terms. So, it is unclear what the conclusions and the strengths of these observations are.
- 4) The points that detected astrocyte subtypes may represent dynamic state is discussed in relation to substates in disease conditions. The effect of neuronal activity on astrocyte substate should be included in this discussion. For instance when discussing Sapkota et al., 2022, in which activity is shown to induce changes in astrocyte transcripts and proteome.

Minor points:

- 1) The abstract is very short, and difficult to understand for non-specialists: i) it lacks an introduction on the subject of astrocytes, ii) the term 'stable subtypes' needs more explanation here, iii) the point 'are detected subtypes actually stable subtypes, or do they represent dynamic states?' is important but not clear from the abstract.
- 2) The authors make a point on diversity of juxtavascular astrocytes. A point that is missed is that astrocytes differ in the interaction with blood vessels, this also different per brain region, see Hosli, L. et al. Direct vascular contact is a hallmark of cerebral astrocytes. Cell Rep 39, 110599 (2022).
- 3) Page 3: remove comma in: '...fibrous, WM astrocytes..'
- 4) The paragraph "Protocol improvements: sequencing workflows" is missing a conclusion on the protocol improvements.
- 5) Page 12: "Sequencing-based spatial methods are high-throughput and can map the whole transcriptome to the tissue, but often lack sensitivity and single-cell resolution." Please provide examples in literature.

Referee #2:

The current review entitled 'Astrocyte diversity and subtypes: Aligning transcriptomics with multimodal perspectives' by Hennes et al provides an overview on the topic of astrocyte heterogeneity as assessed by RNA sequencing approaches, cell morphology and CNS location.

The review addresses a timely question and is of general interest to the field of glial biology, however there are several aspects to this manuscript that require further clarity. Below are suggestions for improving the manuscript.

- Why is it important we know the answer to this question from the authors perspective?
- Define some criteria for distinguishing subtype, for example, draw from neuronal literature. Put this in a box or figure. The clearest example the authors present is white matter astrocytes, yet the text is scattered all over the manuscript. It would be useful to define logical criteria using WM and gray matter astrocytes as a representative example, leaving question marks or blank spots for criteria that have not been sufficiently established.
- Figure 3 - As presented to the uninformed reader, the takeaway from this figure would be that some astrocytes are involved in (i.e.) energy metabolism and other subtypes are involved in synaptic modulation. This is not accurate, as most of the properties in this figures are intrinsic, homeostatic functions of all astrocytes.
- Reactive astrocytes are referred to frequently, in different parts of the manuscript. Consider combining this all in the one section toward the end of the article to separate intrinsic properties in healthy tissue vs. in the context of perturbation. Do the authors consider the response to pathology a defining feature of an astrocyte cell type? If so, consider adding it as relevant subtype distinguishing criteria.
- Is transplantation a defining feature for a unique cell type? Again, drawing on the neuronal field, this approach, is rarely, if ever applied.
- Paragraph 3 of the discussion can be removed. The author states the subtypes should maintain stable characteristics. Yet, the article cited refers to astrocytes substates across disease progression. In no instance would one expect a cell type to remain stable in a rapidly changing disease state.
- There are many instances where investigations have shown functionally distinct properties of astrocytes from different regions. What functional criteria can/have been used?
- The 5th paragraph of the discussion also needs attention. Some of the statements need clarity. For example, what is live imaging going to tell us- calcium dynamics, coupling, responses to neurotransmitter, etc? There is a comment regarding moving WM astrocytes-no reference. Provide context in which real-time imaging would allow the visualization of substate transition.

Referee #3:

The manuscript by Hennes et al discusses astrocyte diversity and subtypes by aligning transcriptomics with multimodal perspectives. The manuscript is divided into two main parts. The first is providing an overview of astrocyte diversity under physiological and pathological conditions, the second part is discussing consideration and protocol improvements for the interpretation of astrocytic transcriptome data. While the second parts is overall ok, the first parts lacks structure and flow.

Here are my comments to the manuscript:

1. The schematics of the review's content is clear and good, however, this reviewer is surprised that the authors did not include "function" as a box on the lower right ("What information is still missing"). Even though, the authors explicitly write in their abstract that conclusions on stable astrocyte subtypes ideally require a functional analysis, this topic is significantly underrepresented in their review and mention only very shortly in their discussion. Here, they write: "An authentic astrocyte subtype should be defined by their specific functional roles rather than exclusively their transcriptome profiles, as previous efforts to delineate astrocyte heterogeneity have struggled to validate these subtypes at the functional level." It would have been insightful to see a discussion about what functionality means and which parameters could be considered functional read outs. The presented paragraph which follows the sentences above is featureless, expect the last sentence in which patch-seq analysis is mentioned.
2. Introduction as well as the description of astrocyte heterogeneity is rather unstructured and difficult to read. Introduction: The authors start to describe astrocyte molecular and functional diversity that is tailored to specific demands of local circuitries, mentioning differential expression levels of Glutamate transporters across brain regions. While it is certainly true that Glt1 and Glast are differentially expressed in different brain regions (and also in different developmental stages), the functional implication of this remains unclear and the chosen example remains arbitrary. This is a good example of how this review is written: a detail of a study is mentioned, but not satisfyingly discussed and included into the broader picture. Instead, the authors jump from paragraph to paragraph, leaving the reader wondering about their meaning. Since some parts are also redundant with what is discussed afterwards in more detail, my suggestion is to significantly revised the introduction according to structure to better work out the purposed of this review.
3. Seminal studies are missed in the review:

Morel et al., 2017; Lin et al., 2017; Karpf et al., 2022; Green et al., 2024; Cain et al., 2023.

4. To better understand the chapter on astrocyte subtypes in disease, the authors should divide it into acute injury and neurodegenerative disease. Again, it is difficult to understand what the authors want to say due to a missing structure, and jumping from one specific finding to another. Why do the authors focus mostly on AD?

The second part of the review on consideration and protocol improvements for the interpretation of astrocytic transcriptome data is valuable and interesting to the astrocyte field. Overall, my suggestion is to substantially revise the first part of this manuscript in order to give it a clearer concept and structure.

POINT-BY-POINT RESPONSE

Referee #1:

This is an interesting and timely review on heterogenous astrocyte gene expression and its meaning for distinct astrocyte subtypes versus substates.

There are a number of points that need improvement.

Response:

Dear Reviewer 1,

We sincerely thank you for your thoughtful and constructive comments on our review manuscript. We greatly appreciate the time and effort you invested in evaluating our work. Below, we provide a detailed point-by-point response outlining how we have addressed each of your comments in the revised version of the manuscript.

1) The text on page 6-7, "In general... " until ' ..are needed (Colonna2024)" is unclear. The purpose and conclusion of this part is not clear and not adding much. It is describing data of Figure 2 but that text and figure is also not clear:

In the section 'Exploring homeostatic astrocyte heterogeneity in the sc/snRNA-seq era' we want to provide an overview of literature regarding the many different astrocyte subtypes that have been identified across brain regions and how they distinguish themselves from one another. Besides how they differ, it is also of interest to the field to comprehend what the real core or pan-astrocyte functions are, which are also rather unexplored. We believe that the strategy implied in Mathys et al 2024, could provide additional information to further unravel this topic and as a proof-of-concept implemented this analytical strategy on our own data. We have rewritten this part of the manuscript and now added a new paragraph about 'Pan-astrocyte functions and the division of labor among subtypes'.

2) Figure 2. This figure should be self-explanatory, with the legend describing: 1) the source of the data, 2) the methods used, and 3) the meaning of lines and symbols. Panels C, D, E: what is the meanin of 'is a', and of 'part of'? Also the red lines of 'part of' are not seen. Mistake in 'SiNgal transduction'. Remove the space in '..using simona ."

We understand your point and have added the requested additional information to the figure legend. Red lines are depicted stronger now and we added information in the Figure legend that they are shown in panel C and E.

3) Figure 2. The pan-astrocyte module 4 was described to reveal typical astrocyte functions such as signal transduction and metabolic processes. How significant is this? Also module 6 and 2 (GM and WM) have terms that could be considered pan-astrocte, including 'steroid hormone' terms. It is not surprising anyway that different modules have different GO terms. So, it is unclear what the conclusions and the strengths of these observations are.

The GO analysis was done in STRING, selecting the top 15 biological processes (BP) terms per module with FDR < 0.05. We have also added this information regarding the significance into the figure legend.

With this analysis we want to highlight that this strategy can reveal what the pan-astrocyte functions are and how they could potentially be performed to a different extent by the different astrocyte subtypes. We have adjusted the text so this point comes across better.

4) The points that detected astrocyte subtypes may represent dynamic state is discussed in relation to substates in disease conditions. The effect of neuronal activity on astrocyte substate should be included in this discussion. For instance when discussing Sapkota et al., 2022, in which activity is shown to induce changes in astrocyte transcripts and proteome.

In our discussion we aimed to put forward that previously identified subtypes are actually dynamic substates. It is true that we reference mostly to ‘substates’ in disease conditions as so far clear evidence for a substate under homeostatic conditions is still missing. In Sapkota et al., they indeed show that neuronal activity can modulate astrocyte transcriptome and proteome. These findings were however obtained with TRAP RNA seq, not scRNA seq, and were found to be a robust astrocyte response across regions and therefore not a dynamic ‘state’. We do agree that if astrocyte substates exist under homeostatic conditions, neuronal activity would play a role in their substate transition and have mentioned this now in the discussion.

Minor points:

1) The abstract is very short, and difficult to understand for non-specialists: i) it lacks an introduction on the subject of astrocytes, ii) the term 'stable subtypes' needs more explanation here, iii) the point 'are detected subtypes actually stable subtypes, or do they represent dynamic states'? is important but not clear from the abstract.

We have adjusted the abstract and incorporated your suggestions.

2) The authors make a point on diversity of juxtavascular astrocytes. A point that is missed is that astrocytes differ in the interaction with bloodvessels, this also different per brain region, see Hosli, L. et al. Direct vascular contact is a hallmark of cerebral astrocytes. Cell Rep 39, 110599 (2022).

We have now included this reference in the introduction as well as in a new paragraph where we discuss what the pan-astrocyte functions are and how distinct astrocyte subtypes might perform them to a different extent, such as BBB maintenance that has been mostly characterized for GM astrocytes and is less evident for WM astrocytes.

3) Page 3: remove comma in: ‘.fibrous, WM astrocytes.’

We have adapted this in the manuscript.

4) The paragraph "Protocol improvements: sequencing workflows" is missing a conclusion on the protocol improvements.

We added a conclusion to this section to indicate that techniques capable of effectively combining the high sensitivity of low-throughput methods with the cellular coverage of high-throughput platforms, would provide more comprehensive transcriptomic information and therefore hold significant potential to advance our understanding of astrocyte heterogeneity under both physiological and pathological conditions.

5) Page 12: "Sequencing-based spatial methods are high-throughput and can map the whole transcriptome to the tissue, but often lack sensitivity and single-cell resolution." Please provide examples in literature.

In order to support this statement, references were added that indicate that often the 'capture area' would contain multiple cells and that the sequencing depths are lower compared to sc/snRNA-seq.

Referee #2:

The current review entitled 'Astrocyte diversity and subtypes: Aligning transcriptomics with multimodal perspectives' by Hennes et al provides an overview on the topic of astrocyte heterogeneity as assessed by RNA sequencing approaches, cell morphology and CNS location.

The review addresses a timely question and is of general interest to the field of glial biology, however there are several aspects to this manuscript that require further clarity. Below are suggestions for improving the manuscript.

Response:

Dear Reviewer 2,

We would like to thank you for your careful review of our manuscript and for your valuable and insightful comments. Your feedback has helped us to improve the clarity and quality of our review. In the following, we provide a detailed response to each of your points and explain the corresponding changes made in the revised version of the manuscript.

- *Why is it important we know the answer to this question from the authors perspective?*

We further clarified in the revised manuscript why it is now particularly timely to overview the transcriptome studies on astrocytes and highlight the issues in aligning them. The point is that the field has moved from considering astrocytes largely similar throughout the brain, to a wealth of heterogeneity claims from sc/snRNAseq. However, these RNA-seq data are often difficult to align with other scRNA-seq studies and often lack functional confirmation. This

leads to a confusion in the field about if and how many astrocyte subtypes there are. We aim to clarify this by introducing the topic of subtype classifications, providing a comprehensive overview about previous astrocyte transcriptome studies and the different technologies used, including a discussion on their respective advantages and disadvantages, before calling for a code of conduct for claims on astrocyte subtypes.

In our view this is an important contribution to the field, as astrocytes perform many vital tasks in the CNS to support neuronal functioning and brain homeostasis in health and disease. However, compared to neurons, they remain greatly understudied and a better comprehension of their biology, including their subtype composition and heterogeneous dynamic nature, could provide useful insights to better understand brain functioning and reveal potential new therapeutic strategies to target neurological disorders. We have now included this at the end of the discussion.

• Define some criteria for distinguishing subtype, for example, draw from neuronal literature. Put this in a box or figure. The clearest example the authors present is white matter astrocytes, yet the text is scattered all over the manuscript. It would be useful to define logical criteria using WM and gray matter astrocytes as a representative example, leaving question marks or blank spots for criteria that have not been sufficiently established.

This is an excellent suggestion that we implemented now by starting with a Figure 1 describing the main criteria for neuronal subtypes and directly aligning them with possible analogous criteria for astrocytes. We discuss this in a preamble as well as the introduction and also come back to this comparison in the discussion. This indeed helps a lot to clarify the points of subtype characterization.

• Figure 3 - As presented to the uninformed reader, the takeaway from this figure would be that some astrocytes are involved in (i.e.) energy metabolism and other subtypes are involved in synaptic modulation. This is not accurate, as most of the properties in this figures are intrinsic, homeostatic functions of all astrocytes.

Indeed, the functions represented in Figure 3 are pan-astrocyte functions. We have revised the figure legend to make this, as well as our interpretation of subtypes/substates, more clear.

• Reactive astrocytes are referred to frequently, in different parts of the manuscript. Consider combining this all in the one section toward the end of the article to separate intrinsic properties in healthy tissue vs. in the context of perturbation. Do the authors consider the response to pathology a defining feature of an astrocyte cell type? If so, consider adding it as relevant subtype distinguishing criteria.

This is also a great suggestion that we followed now by restructuring the review to first describe what is known about astrocyte heterogeneity in the intact, homeostatic condition and then after acute versus chronic brain pathology.

- *Is transplantation a defining feature for a unique cell type? Again, drawing on the neuronal field, this approach, is rarely, if ever applied.*

By definition, cell intrinsic features can be detected only if the extrinsic environment is changed. Therefore it is our opinion that the ultimate validation of a true subtype could be retrieved using transplantation in order to confirm that their core subtype characteristics remain stable in a different environment. However, we acknowledge that this would pose significant technical challenges and may not be feasible with current methodologies. We have therefore removed it from the discussion and placed it in our ‘Box of questions to be further pursued in the field of astrocyte heterogeneity’.

- *Paragraph 3 of the discussion can be removed. The author states the subtypes should maintain stable characteristics. Yet, the article cited refers to astrocytes substates across disease progression. In no instance would one expect a cell type to remain stable in a rapidly changing disease state.*

Initial studies from the Barres group, exploring astrocyte response to injury, stated the appearance of two astrocyte subtypes; the A1 ‘neurotoxic’ and A2 ‘neuroprotective subtype’. While this binary framework provided an important foundation for understanding astrocyte responses to injury, it is now widely recognized as an oversimplification, as articulated in the consensus paper by Escartin et al. However, Zhang et al recently demonstrated that these previously identified neurotoxic and neuroprotective astrocyte subtypes are actually both substates of the exact same cell. This is a novel finding in the field and important for considerations regarding astrocyte subtype vs substate. Furthermore it also supports our recommendation to perform longitudinal analysis for understanding astrocyte heterogeneity. We have thus retained this paragraph in the discussion but revised the text to better reflect the current understanding and relevance of this distinction.

- *There are many instances where investigations have shown functionally distinct properties of astrocytes from different regions. What functional criteria can/have been used?*

We have added an additional paragraph ‘Functional validation of molecularly identified astrocyte subtypes’ to address what characterization has been conducted to address functional diversity between subtypes, such as exploring Ca-signals or the proliferation of astrocytes. We further touch on how functional analysis of other aspects, e.g. metabolism, may be complicated by the lack of available technologies or genetic/immunohistological tools for the subtypes identified within a region. In addition we also propose an alternative strategy to investigate functional differences between subtypes.

- *The 5th paragraph of the discussion also needs attention. Some of the statements need clarity. For example, what is live imaging going to tell us- calcium dynamics, coupling, responses to neurotransmitter, etc? There is a comment regarding moving WM astrocytes-no*

reference. Provide context in which real-time imaging would allow the visualization of substate transition.

We agree that migration of astrocytes in adult brain is rather unexplored/validated so we adapted the text and suggest that live imaging could potentially allow the visualisation of substate transitions that could be accompanied by changes in calcium dynamics, blood vessel association or synaptic coverage.

Referee #3:

The manuscript by Hennes et al discusses astrocyte diversity and subtypes by aligning transcriptomics with multimodal perspectives. The manuscript is divided into two main parts. The first is providing an overview of astrocyte diversity under physiological and pathological conditions, the second part is discussing consideration and protocol improvements for the interpretation of astrocytic transcriptome data. While the second parts is overall ok, the first parts lacks structure and flow.

Response:

Thank you for your thoughtful review and for your positive feedback on the second part of our manuscript. We have revised the first part to improve its structure and flow in order to enhance clarity and readability. We also added a new Figure 1 and a Preamble to outline the main question based on how neuronal subtypes are distinguished. Below, we provide detailed responses to each of your comments and describe how we have addressed them in the revised manuscript.

1. The schematics of the review's content is clear and good, however, this reviewer is surprised that the authors did not include "function" as a box on the lower right ("What information is still missing"). Even though, the authors explicitly write in their abstract that conclusions on stable astrocyte subtypes ideally require a functional analysis, this topic is significantly underrepresented in their review and mention only very shortly in their discussion. Here, they write: "An authentic astrocyte subtype should be defined by their specific functional roles rather than exclusively their transcriptome profiles, as previous efforts to delineate astrocyte heterogeneity have struggled to validate these subtypes at the functional level." It would have been insightful to see a discussion about what functionality means and which parameters could be considered functional read outs. The presented paragraph which follows the sentences above is featureless, expect the last sentence in which patch-seq analysis is mentioned.

We agree with reviewer 3 that this part was indeed missing in our manuscript. We have added an additional paragraph 'Functional validation of molecularly identified astrocyte subtypes' to describe previous functional characterizations and to which extent functional diversity between subtypes has been demonstrated, e.g. between WM and GM astrocytes. We furthermore discuss the challenges to assess functions for which genetic/immunohistological tools may still be missing and also propose an alternative strategy to investigate functional differences between subtypes.

2. Introduction as well as the description of astrocyte heterogeneity is rather unstructured and difficult to read.

Introduction: The authors start to describe astrocyte molecular and functional diversity that is tailored to specific demands of local circuitries, mentioning differential expression levels of Glutamate transporters across brain regions. While it is certainly true that Glt1 and Glast are differentially expressed in different brain regions (and also in different developmental stages), the functional implication of this remains unclear and the chosen example remains arbitrary. This is a good example of how this review is written: a detail of a study is mentioned, but not satisfyingly discussed and included into the broader picture. Instead, the authors jump from paragraph to paragraph, leaving the reader wondering about their meaning. Since some parts are also redundant with what is discussed afterwards in more detail, my suggestion is to significantly revised the introduction according to structure to better work out the purposed of this review.

As suggested we have revised the introduction to make it more concise and restructured the text of the first part of the manuscript also adding more subtitles to improve the logic, flow and readability. In addition, we added a Preamble to outline more clearly the topic of subtype definitions based on neuronal subtype definitions for which we also added a new Figure 1.

3. Seminal studies are missed in the review:

Morel et al., 2017; Lin et al., 2017; Karpf et al., 2022; Green et al., 2024; Cain et al., 2023.

Thank you for bringing this to our attention, these papers have now been included in the manuscript.

4. To better understand the chapter on astrocyte subtypes in disease, the authors should divide it into acute injury and neurodegenerative disease. Again, it is difficult to understand what the authors want to say due to a missing structure, and jumping from one specific finding to another. Why do the authors focus mostly on AD?

This is an excellent suggestion. We have now revised this part of the text and separated acute from chronic pathology.

Prof. Magdalena Götz
Helmholtz Munich
Institute of Stem cell research
Ingolstädter Landstraße 1
München, Bavaria 85764
Germany

Dear Magdalena,

I am pleased to inform you that your review has been accepted for publication in EMBO reports. Your manuscript will be processed for publication by EMBO Press. It will be copy edited and you will receive page proofs prior to publication. Please note that you will be contacted by Springer Nature Author Services to complete licensing information.

Please use this token when signing the license agreement: Unavailable.
So that the publication charges for this review will be waived.

All the best,
Esther
